Identification of L-asparaginases from Streptomyces strains with competitive activity and immunogenic profiles: a bioinformatic approach

González-Torres Iván 1
http://orcid.org/0000-0002-6879-0673 Perez-Rueda Ernesto 2
http://orcid.org/0000-0003-3145-0824 Evangelista-Martínez Zahaed 3
Zárate-Romero Andrés 1 4
Moreno-Enríquez Angélica 5
http://orcid.org/0000-0002-0156-6773 Huerta-Saquero Alejandro 1 saquero@cnyn.unam.mx
1 Centro de Nanociencias y Nanotecnología, Universidad Nacional Autónoma de México , Ensenada, Baja California , México
2 Instituto de Matemáticas Aplicadas y Sistemas, Universidad Nacional Autónoma de México , Mérida, Yucatán , México
3 Subsede Sureste, Centro de Investigación y Asistencia en Tecnología y Diseño del Estado de Jalisco, AC , Mérida, Yucatán , México
4 Consejo Nacional de Ciencia y Tecnología , Ciudad de México , Mexico
5 Unidad Genómica Metabólica, Universidad Marista de Mérida , Mérida, Yucatán , México
Gillespie Joseph
Electronic publication date: 2020 Nov 13
Publication date: 2020
Volume: 8
Electronic Location ID: e10276
Received 2020 Apr 3; Accepted 2020 Oct 8
Copyright: © 2020 González-Torres et al.
Copyright year: 2020
Copyright holder: González-Torres et al.
License: This is an open access article distributed under the terms of the Creative Commons Attribution License, which permits unrestricted use, distribution, reproduction and adaptation in any medium and for any purpose provided that it is properly attributed. For attribution, the original author(s), title, publication source (PeerJ) and either DOI or URL of the article must be cited.
License URL: https://creativecommons.org/licenses/by/4.0/

Keywords: L-asparaginase, Acute lymphocytic leukemia, Streptomyces, Bioinformatics

Funding: PAPIIT-DGAPA UNAM IN210618 and IN209620 Programa Iberoamericano de Ciencia y Tecnología para el Desarrollo P918PTE0261 This work was supported by PAPIIT-DGAPA UNAM grant IN210618 (Alejandro Huerta-Saquero) and grant IN209620 (Ernesto Perez-Rueda); and Programa Iberoamericano de Ciencia y Tecnología para el Desarrollo (P918PTE0261) (Ernesto Perez-Rueda). The funders had no role in study design, data collection and analysis, decision to publish, or preparation of the manuscript.

==============================
The enzyme L-asparaginase from Escherichia coli is a therapeutic enzyme that has been a cornerstone in the clinical treatment of acute lymphoblastic leukemia for the last decades. However, treatment effectiveness is limited by the highly immunogenic nature of the protein and its cross-reactivity towards L-glutamine. In this work, a bioinformatic approach was used to identify, select and computationally characterize L-asparaginases from Streptomyces through sequence-based screening analyses, immunoinformatics, homology modeling, and molecular docking studies. Based on its predicted low immunogenicity and excellent enzymatic activity, we selected a previously uncharacterized L-asparaginase from Streptomyces scabrisporus. Furthermore, two putative asparaginase binding sites were identified and a 3D model is proposed. These promising features allow us to propose L-asparaginase from S. scabrisporus as an alternative for the treatment of acute lymphocytic leukemia.

Introduction

Acute lymphocytic leukemia (ALL) is a hematological disorder of the bone marrow and is characterized by abnormal proliferation of immature lymphoid line cells, blocked at an early stage of cell differentiation, that accumulate and replace healthy hematopoietic cells in the bone marrow (Pui, Relling & Downing, 2004; Onciu, 2009). ALL occurs predominantly in children of 1–4 years of age and represents approximately 25% of childhood cancers and about 80% of leukemias (Katz et al., 2015).

Although in most cases the risk factors and pathogenicity associated with ALL have not been clearly identified, the etiology of the disease has been mainly associated with a variety of conditions; cytogenetic alterations, mutations to key genes that regulate cellular proliferation, differentiation and death; presence of oncogenic viruses, immunodeficiency, exposure to pesticides, solvents, and ionizing radiation (Pui, Relling & Downing, 2004; Bassan, Maino & Cortelazzo, 2016).

Treatment for ALL patients involve steroid drugs, prednisone, vincristine, and the enzyme L-asparaginase (ASNase) (Avramis, 2012; Schwab & Harrison, 2018). ASNase has been essential in the treatment of ALL since the 1970s, with demonstrated effectiveness as an individual drug with remission rates of up to 68% (Salzer et al., 2017). The combination of ASNase with other anticancer drugs has led to remission rates of up to 90% (Lanvers-Kaminsky, 2017).

Currently, there are four ASNase formulations available for the ALL treatment: two native forms of the enzyme, obtained from Escherichia coli (EcAII) and Erwinia chrysanthemi (ErAII), and pegylated E. coli ASNase (EcAII-PEG), as well as pegylated E. chrysanthemi ASNase (ErAII-PEG). Of these, EcAII-PEG has become the first-line treatments for ALL in the US, with EcAII the most widely used formulation. ErAII is administered to patients who have developed hypersensitivity to the above formulations (Pieters et al., 2011; Abribat, 2016; Barba et al., 2017). In recent years, evidence has been accumulating of its usefulness as an important component in the treatment of other hematological malignancies, such as acute myeloid leukemia, myelosarcoma, lymphosarcoma, Hodgkin’s disease, and chronic lymphocytic leukemia (Emadi, Zokaee & Sausville, 2014; Lopes et al., 2015). Despite their high antileukemic potential, the use of ASNases in the treatment of ALL is limited by their toxicity. Among the adverse effects that have been reported are leukopenia, immune suppression, acute pancreatitis, liver dysfunction, hyperglycemia, abnormalities in hemostasis, and hemorrhages of the central nervous system (Schein et al., 1969; Ramya et al., 2012; Chan et al., 2014; Ali et al., 2016; Hijiya & Van der Sluis, 2016; Kamal et al., 2019).

The generation of immune responses during treatment with ASNase is a common condition that has been reported in up to 75% of patients. These reactions depend on the formulation used, the mode of administration (intravenous or intramuscular), and the treatment protocol (Hijiya & Van der Sluis, 2015). For example, between 30% and 75% of patients that receive the native form of the E. coli enzyme experience hypersensitivity reactions, and about 70% develop anti-EcAII antibodies after drug administration (Battistel et al., 2020); these antibodies lead to rapid inactivation of the enzyme (Walenciak et al., 2019).

Allergic reactions to ASNase, which are associated with its bacterial origin, range from mild urticaria to life-threatening anaphylactic shock. Irritation, fever, vomiting, gastrointestinal edema, and breathing difficulties are symptoms frequently reported (Lanvers-Kaminsky, 2017). On the other hand, adverse effects have been reported due to the toxicity produced by glutaminase cross activity, such as leukopenia, immune suppression, acute pancreatitis, hyperglycemia, thrombosis, neurotoxicity, and liver failure, among others (Ramya et al., 2012; Chan et al., 2014; Ali et al., 2016).

Different strategies to reduce the toxicity of ASNase have been reported, including modifications in the structure of the protein by mutagenesis, design of mutants with diminished ability to hydrolyze L-glutamine, chemical modifications in specific amino acids, and modifications to drug formulations (Ramya et al., 2012; Nguyen, Su & Lavie, 2016; Nguyen et al., 2018). Covalent conjugation of the enzyme with polyethylene glycol, known as PEGylation, reduces the incidence of hyperglycemia, pancreatitis, and anaphylaxis. This specific modification increases the half-life of the enzyme and reduces the frequency of drug administration (Thomas & Le Jeune, 2016).

On the other hand, the exploration of new sources of ASNases offers the possibility of finding versions of the enzyme with different pharmacological characteristics, potentially useful for the treatment of ALL and other lymphomas (Krishnapura, Belur & Subramanya, 2016). In this sense, besides searching for less immunogenic asparaginases, it is essential to find those with high affinity for L-asparagine (in the micromolar range) in order to have the potential for therapeutic use. Some atypical ASNases, unrelated to EcAII and ErAII, such as Rhizobium etli type II ASNase (ReAII) (Ortuño-Olea & Durán-Vargas, 2000), have been proposed as alternatives with therapeutic potential; the R. etli ASNase has null glutaminase activity and a different immunogenic profile than E. coli and E. chrysantemy ASNases (Moreno-Enriquez et al., 2012; Huerta-Saquero et al., 2013). However, this enzyme has a low affinity for asparagine, which limits its use. Despite the success of E. coli and E. chrysanthemi ASNases in therapeutic regimens for ALL and other types of leukemia, the search for new ASNases that are less toxic and less immunogenic is necessary. In this sense, ASNases from phylogenetically distant microorganisms offer a specific target for the selection of variants with the appropriate characteristics. Among these, ASNases from Streptomyces are one potential group to be evaluated for immunogenicity, toxicity, and affinity for L-asparagin to obtain new ASNases with therapeutic potential. The main characteristics to select asparaginases with therapeutic potential are the high affinity for L-asparagine (in the micromolar range), null or low glutaminase cross-activity, as well as a different three-dimensional folding from the E. coli asparaginase, which suggests different immunogenicity. In this work, we develop a strategy based on bioinformatics tools to analyze and select ASNases from Streptomyces for ALL treatment, taking advantage of its phylogenetic distance from E. coli, looking for those candidates that meet the two fundamental criteria: asparaginases with high affinity for asparagine (using active site prediction tools and molecular docking), and that have lower immunogenicity (using antigenicity and protein structure prediction tools). As a reference, we selected the E. coli and Streptomyces coelicolor ASNases. The importance of this novel approach is discussed.

Materials and Methods

Identification and selection of homologous L-Asparaginases

Putative ASNases from Streptomyces were identified through a BLASTp search against the NR database of the NCBI using as seeds the amino acid sequences of EcAII (ID P00805) and Streptomyces coelicolor type II ASNase (ScAII; ID Q9K4F5). The search was restricted to the Streptomyces taxon (Taxonomic ID number: 1883), and an E-value less than 1e−06 was considered significant. Partial proteins and those from unidentified Streptomyces strains were excluded. In a posterior step, the set of protein sequences was filtered at 60% identity as cutoff to avoid redundancy, using the CD-Hit program (http://weizhongli-lab.org/cdhit_suite/cgi-bin/index.cgi) (Huang et al., 2010). Each cluster was analyzed using the HMMER program on the PFAM server (http://pfam.xfam.org/) to determine the protein family to which they belonged (Finn, Clements & Eddy, 2011; Finn et al., 2016).

Phylogenetic analysis

ASNases amino acid sequence alignments were performed using Clustal Omega (Sievers et al., 2011) with default parameters. The quality of the alignments was improved using the model PF06089.11 or PF00710.11 of ASNase, as required. Multiple sequence alignment statistics were computed with AliStat (http://www.csb.yale.edu/userguides/seq/hmmer/docs/node27.html).

Phylogenetic analyses were carried out using the maximum-likelihood method with the program Mega 7. The WAG model was chosen as substitution model, and 1,000 replicates were performed. The best tree was calculated using the majority rule. Additionally, E. coli type I ASNase (EcAI) was included in the phylogenetic analysis of the PF00710.11 cluster. EcAI is closely related to EcAII but it does not have therapeutic potential. For the PF06089.11 cluster, Rhizobium etli type II ASNase (ReAII) was included in the analysis.

Antigenicity prediction

The prediction of the probability of antigenicity of each ASNase was calculated with the server ANTIGENpro (http://scratch.proteomics.ics.uci.edu/) (Magnan et al., 2010). ANTIGENpro is a sequence-based, alignment-free, protein antigenicity predictor with an estimated accuracy of 82%.

HLA class II binding prediction

The amino acid sequence of each candidate ASNase was screened for T-cells epitopes with the MHC II Analysis Resource at the Immune Epitope Data Base (IEDB) server (http://tools.iedb.org/mhcii/). MHC II Analysis Resource parses sequences into 15-mer and assesses the binding potential of each 15-mer to MHC class II molecules of one or more HLA alleles. The IEDB recommended method was used for predictions for a set of 8 HLA alleles that collectively represent >95% world population: HLA-DRB1*01:01, HLA-DRB1*03:01, HLA-DRB1*04:01, HLA-DRB1*07:01, HLA-DRB1*08:01, HLA-DRB1*11:01, HLA-DRB1*13:01 and HLA-DRB1*15:01. The IEDB-recommended method uses the consensus approach, combining NN-align, SMM-align, CombLib, Sturniolo, and NetMHCIIpan (Wang et al., 2010). For each peptide, a percentile rank is generated by comparing the peptide’s score against the scores of five million random 15-mer selected from SWISSPROT database, and the median percentile rank is used to calculate a consensus percentile rank (CPR). Peptides with a CPR < 2 were defined as high-affinity binders and thus selected for epitope density (ED) calculation. Multiple 9-mer cores were identified in overlapped 15-mer peptides. To reduce overestimation of predicted peptides, only the 9-mer cores, predicted by using the Sturniolo method (Sturniolo et al., 1999) and with a CPR < 1, were considered for the analysis. Finally, epitope density (ED) was calculated using the follow equation, modified from (Santos et al., 2013): ED=Predictedepitope∗(2−Affinityaverage(cpr))Proteinlengthsize−Epitopesize+1

Where Predicted epitope is the number of epitopes with a CPR < 1.

Epitope coverage was calculated as the number of alleles covered by the epitope consensus, according to the following assumption: when a small number of alleles is covered, a lower percentage of the population will develop sensitivity to ASNase.

Protein structure prediction, refinement and quality assessment

The three-dimensional structures of the selected ASNases was modeled by homology using the I-Tasser server (https://zhanglab.ccmb.med.umich.edu/I-TASSER/) (Zhang, 2008). In brief, starting from an amino acid sequence, I-Tasser generates three-dimensional atomic models from multiple threading alignments and iterative structural assembly simulations. A C-score, provided as an estimate of the accuracy of the models generated, typically ranges between −5 and +2, with a higher value indicating higher confidence, and vice versa (Roy, Kucukural & Zhang, 2010).

For each ASNase, the model with the higher C-score was selected and then refined using the ModRefiner server (https://zhanglab.ccmb.med.umich.edu/ModRefiner/). ModRefiner improves the physical quality and structural accuracy of three-dimensional protein structures by a two-step, atomic-level energy minimization (Xu & Zhang, 2011). Finally, the quality of the models was evaluated by PROCHECK (https://servicesn.mbi.ucla.edu/PROCHECK), Qmean (https://swissmodel.expasy.org/qmean/), and Verify3D (http:/servicesn.mbi.ucla.edu/Verify3D).

Molecular docking

The molecular coupling was carried out using Autodock Tools software (Sanner, 1999; Morris et al., 2009). EcAII (PDB ID: 3ECA) was recovered from the PDB protein database (http://www.rcsb.org/) (Swain et al., 1993; Berman et al., 2000). Once refined, selected ASNase structures were prepared using Dock prep at UCSF Chimera and refined using the Gasteiger method (Gasteiger & Marsili, 1978).

The three-dimensional structures of the asparagine and glutamine ligands were obtained from the DrugBank repository (https://www.drugbank.ca/; accession numbers DB00174 and DB00130, respectively) (Wishart et al., 2018). The preparation of the ligands was carried out by the Gasteiger method and, finally, the allocation of the rotation centers was determined (Gasteiger & Marsili, 1978).

For each ASNase, the search box was focused on previously proposed active sites. The box size was defined to cover all residues of the ligand binding site, using a grid size of 0.375 Å.

Blind molecular docking was performed with Autodock 4.2 software, using the Lamarckian genetic algorithm, with 1,000 runs, for a population size equal to 150, with 2.5 × 106 evaluations, a mutation rate equal to 0.02 in 27,000 generations.

In addition, the active site location was predicted by AutoLigand (Harris, Olson & Goodsell, 2008). Briefly, AutoLigand identifies sites of maximum affinity from maps generated by AutoGrid, finding regions with better energy and a lower volume.

Results

L-Asparaginases from Streptomyces cluster into two type families according to its protein architecture

The Blast search against the Streptomyces taxon revealed 296 putative ASNases homologous to EcAII and 703 homologous to ScAII with a significant score. After manual examination of both groups, 136 and 311 complete sequences were kept for EcAII and ScAII groups, respectively. Protein domain analysis using PFAM server showed that 136 sequences are related to the PF00710.11 family of N-terminal ASNases. For sequences homologous to ScAII, PFAM analysis revealed that they belong to the PF06089.11 family of ASNases, a group of enzymes related to ReAII, a thermolabile enzyme induced by L-asparagine and repressed by the carbon source (Moreno-Enriquez et al., 2012; Huerta-Saquero et al., 2013). Representative clusters for PF00710.11 and PF06089.11 families obtained using the CD-Hit suite program were generated at a 60% identity cutoff, with 19 and 7 putative ASNases, respectively (Table 1). ASNases sequences showed similar lengths in both clusters, ranging from 320 to 420 amino acids.

Table 1 Representative Streptomyces ASNases of the PF00710.11 and PF06089.11 families, at 60% identity cutoff.

ASNase ID	Organism	Length
(amino acids)	Family	
PF00710.11 family	
WP_053609500.1	S. purpurogeneiscleroticus	373	PF00710.11	
WP_053610569.1	S. purpurogeneiscleroticus	338	PF00710.11	
WP_055617501.1	S. phaeochromogenes	380	PF00710.11	
WP_051815467.1	S. lavenduligriseus	363	PF00710.11	
WP_078649241.1	S. fradiae	350	PF00710.11	
EFL23513.1	S. himastatinicu ATCC 53653	351	PF00710.11	
WP_014151616.1	S. cattleya	331	PF00710.11	
WP_095730579.1	S. albidoflavus	333	PF00710.11	
WP_078965752.1	S. aureocirculatus	343	PF00710.11	
WP_078513220.1	S. purpureus	421	PF00710.11	
WP_009718687.1	S. himastatinicus	347	PF00710.11	
WP_079189481.1	S. paucisporeus	384	PF00710.11	
WP_052425051.1	S. fulvoviolaceus	340	PF00710.11	
ELP65653.1	S. turgidiscabies Car8	358	PF00710.11	
WP_070201703.1	S. nanshensis	347	PF00710.11	
KWW98572.1	S. thermoautotrophicus	333	PF00710.11	
WP_073950513.1	S. kebangsaanensis	333	PF00710.11	
WP_030748190.1	S. griseus	329	PF00710.11	
WP_059134811.1	S. alboniger	332	PF00710.11	
PF06089.11 family	
ARZ68596.1	S. albireticuli	428	PF06089.11	
CDR15801.1	S. iranensis	387	PF06089.11	
SOD64826.1	S. zhaozhouensis	316	PF06089.11	
WP_020554088	S. scabrisporus	332	PF06089.11	
WP_044373749	S. ahygroscopicus	330	PF06089.11	
WP_078645645	S. varsoviensis	348	PF06089.11	
WP_078980718.1	S. scabrisporus	327	PF06089.11	

The sequences belonging to the PF00710.11 family have conserved residues located at the ligand binding site necessary for L-asparagine hydrolysis: Thr 12, Tyr 25, Ser 58, Gln 59, Thr 89, Asp 90, and Lys 162 for subunit A; Asn 248 and Glu 283 for subunit C. In this regard, Thr 12–Lys 162–Asp 90 and Thr 12–Tyr 2–Glu 283 are the catalytic triads involved in L-asparagine hydrolysis, where Thr 12 and Thr 89 are involved in the nucleophilic attack of the substrate (Gesto et al., 2013; Sanches, Kraunchenko & Polikarpov, 2016).

Concerning the PF06089.11 family, we identified an N-terminal conserved motif, with sequences NCSGKHxAM, DGCGAPL, SHSGEx(2)H, and PRSx(2)KPxQ probably involved in asparagine hydrolysis. ReAII hydrolyzes L-asparagine at similar levels to Erwinia chrysanthemi, but with lower affinity than L-asparaginases from both E. coli and E. chrysanthemi (Moreno-Enriquez et al., 2012). Furthermore, ReAII is the only ASNase characterized from the PF06089.11 family.

Phylogenetic analysis of ASNases

For the PF00710.11 family, EcAI was added to the multiple sequence alignment in order to know the relationship between this ASNase and the candidate ASNases. EcAI belongs to the same family of proteins as EcAII, but it does not represent a therapeutic option for ALL treatment. It is noteworthy that asparaginases can also be classified according to subcellular localization, (a) periplasmic asparaginases containing secretion signal peptide and, (b) asparaginases with intracellular localization. The former generally have a higher affinity for asparagine. However, according to their architecture, both types of proteins can be found in the PF00710.11 or PF06089.11 families. This is the case of E. coli asparaginases I and II, both belonging to the PF00710.11 family (https://pfam.xfam.org/family/PF00710#tabview=tab1). We found that the ASNase with accession number WP_059134811.1 of Streptomyces alboniger is grouped in the same clade as EcAI, and so it was excluded from subsequent analyses (Fig. 1A).

Figure 1 Phylogenetic tree of PF00710.11 (A) and PF06089.11 (B) families.

Blue dots highlight reference sequences added to each analysis. Red dots highlight sequences used as internal controls (asparaginases from E. coli and R. etli, respectively). A total of 1,000 replicates were performed. Bootstrap values are indicated.

The phylogenetic reconstruction showed three well-defined clades (Fig. 1A). The first clade includes ASNases from Streptomyces species S. aureocirculatus (WP_078965752.1), S. cattleya (WP_014151616.1), S. thermoautotrophicus (KWW98572.1), S. himastatinicu (EFL23513.1), S. turgidiscabies (ELP65653.1), S. nanshensis (WP_070201703.1), and S. griseus (WP_030748190.1).

The second clade includes ASNases from S. albidoflavus (WP_095730579.1), S. kebangsaanensis (WP_073950513.1), S. fradiae (WP_078649241.1), S. himastatinicus (WP_009718687.1), S. purpureus (WP_078513220.1), and S. paucisporeus (WP_079189481.1). Finally, the third clade contains proteins from S.purpurogeneiscleroticus (WP_053609500.1), S. purpurogeneiscleroticus (WP_053610569.1), S. phaeochromogenes (WP_055617501.1), and S. lavenduligriseus (WP_051815467.1) where EcAII was included, suggesting that proteins clustered in this clade share similar properties to EcAII. In addition, two proteins, WP_053609500.1 and WP_055617501.1, exhibited the largest proportion of antigenic regions, with almost the same probability regions as the EcAII protein.

On the other hand, for the ASNases of PF06089.11, phylogenetic analysis included both the ASNase sequence of R. etli and S. coelicolor (ReAII and ScAII, respectively) (Fig. 1B). The tree defines two clades. In the first one, where the ScAII was included, we also considered ARZ68596.1 from S. albireticuli, SOD64826.1 from S. zhaozhouensis, WP_078645645 from S. varsoviensis, CDR15801.1 from S. iranensis, and WP_020554088 from S. scabrisporus. In the second clade were included the following proteins: WP_044373749 from S. ahygroscopicus and WP_078980718.1 from S. scabrisporus.

Antigenicity predictions

The results for antigenicity showed a likelihood of being antigenic for all ASNases in both sets that was lower than that of EcAII (Fig. 2). Nevertheless, among selected Streptomyces ASNases, the candidates from S. purpurogeneiscleroticus (WP_053609500.1) and S. phaeochromogenes (WP_055617501.1) showed a higher probability of being antigenic, whereas the rest of the ASNases showed very low antigenicity values in comparison with an E. coli ASNase (P00805_EcAII).

Figure 2 ASNase antigenicity predictions.

The antigenicity scores for PF00710.11 family (A) and PF06089.11 family (B) of Streptomyces asparaginases were compared with the EcAII antigenicity score.

T-cell epitope analysis

After antigenicity prediction, the ED, the total number of high-affinity epitopes, the affinity epitopes, and the number of HLA alleles covered by each ASNase were calculated. The results showed that the ASNases with accession numbers WP_053609500.1, WP_053610569.1, EFL23513.1, WP_095730579.1, WP_078513220.1, and WP_052425051.1 have higher EDs than the reference (P00805_EcAII; ED=0.01114; 5 covered alleles) (Fig. 3).

Figure 3 Epitope mapping of ASNases of the PF familes evaluated, (A) PF00710.11 and (B) PF06089.11.

The epitopes identified along with the ASNase sequences are shown. The color intensity represents the number of hits for each of them.

On the other hand, the ASNase with the lowest predicted ED was WP_044373749.1, with an ED of 0.0027 and a coverage of 4 alleles, following by WP_095730579.1 (2 alleles), ELP65653.1 (3 alleles), and Q9K4F5 (3 alleles) (Table 2).

Table 2 High-affinity epitope prediction.

Epitope number, CPR value, allele coverage, and ED of ASNases.

ASNase ID	Epitope number	CPR value	Allele number	ED	
P00805_EcAII	10	0.6383	5	0.0114	
WP_053609500.1	12	0.5174	5	0.0171	
WP_053610569.1	14	0.5381	8	0.0196	
WP_055617501.1	7	0.4532	7	0.0112	
WP_051815467.1	6	0.6673	5	0.0060	
WP_078649241.1	7	0.4554	6	0.0111	
EFL23513.1	10	0.6054	8	0.0115	
WP_014151616.1	3	0.4024	5	0.0056	
WP_095730579.1	8	0.5346	2	0.0115	
WP_078965752.1	3	0.4987	4	0.0045	
WP_078513220.1	9	0.4551	6	0.0119	
WP_009718687.1	5	0.6480	4	0.0052	
WP_079189481.1	4	0.5217	5	0.0051	
WP_052425051.1	10	0.4369	6	0.0170	
ELP65653.1	5	0.6717	3	0.0047	
WP_070201703.1	7	0.6637	6	0.0069	
KWW98572.1	4	0.7254	4	0.0034	
WP_073950513.1	6	0.7424	4	0.0048	
WP_030748190.1	3	0.5125	6	0.0046	
Q9K4F5_ScAII	3	0.4167	3	0.0053	
ARZ68596.1	5	0.6283	5	0.0044	
SOD64826.1	5	0.5046	5	0.0080	
WP_078645645.1	7	0.6404	5	0.0074	
CDR15801.1	5	0.5510	5	0.0059	
WP_078980718.1	6	0.7003	6	0.0056	
WP_044373749.1	3	0.7114	4	0.0027	
Note:

Epitope number refers to the number of epitopes with CPR < 1. Allele number is the number of allele coverage for high affinity epitopes (with a CPR < 1).

Additionally, the distribution of epitopes was mapped into the sequences of the ASNases (Fig. 3). ASNases of the PF06089.11 family tended to have a lower ED (Table 3) as well as lower allele coverage than those of the PF00710.11 family (Fig. 3).

Table 3 SsAII-2 putative binding site residues.

PF06089.11 family conserved residues are shown in bold.

Site	Ligand-binding positions predicted by AutoLigand	Ligand-binding positions predicted by blind molecular docking	
A	Arg 58, Ser 59, Lys 62, Asn 141, Ser 143, Gly 144, Lys 145, His 146, Ala 147, Gly 236, Gly 237, Asp 238, Gly 239, Lys 255, Gly 256, Gly 257, Ala 258, Pro 281, Leu 326	Asn 135, Thr 136, Arg 137, Arg 139, Asn 141, Gly 144, His 146, Asp 192	
B	Ala 85, Gly 86, Ser 87, His 88, Thr 89, Gly 90, Gln 91, His 94, Leu 164, Asp 165, Pro 166, Gly 167, His 168, Leu 173, Glu 177, Gly 178, Asp 180	–	

Next, ASNases with lower allele coverage, lower ED, and lower probability of antigenicity were selected for further analysis. S. coelicolor (Q9K4F5), S. scabrisporus (WP_078980718.1), and S. albireticuli (ARZ68596.1) were selected as promising enzymes.

Protein structure predictions

From selected ASNases, homology-based models were generated (I-Tasser). For the subsequent analysis, the S. scabrisporus asparaginase II model, which had the highest C-value, was chosen (WP_078980718.1 SsAII-2) (Fig. 4). The structural model obtained by I-Tasser (with a C-value of −3.09) was refined with ModRefiner. In addition, the RAMPAGE program (http://mordred.bioc.cam.ac.uk/˜rapper) and Verify3D were used to validate the stereochemical quality of the resulting three-dimensional model. After analyzing the Ramachandran plot, 91.7% and 5.5% of the residues were located in favored and allowed regions, respectively; whereas Verify3D analysis revealed that 80.73% of the residues had an average 3D-1D score <0.2, indicating that the model is compatible with its sequence.

Figure 4 3D protein structure prediction of S. scabrisporus asparaginase II (WP_078980718.1; SsAII-2).

Based on the predicted structure, ASNase WP_0789718.1 (PF06089.11 family) is related in terms of folding to the beta-lactamase family. Beta-lactamases (SCOP data base, entry 56600) consist of a cluster of alpha-helices and an alpha/beta sandwich. This folding is also found in transpeptidases, esterases, penicillin receptors, D-aminopeptidases, and glutaminases (InterPro IPR012338).

Active site prediction

In order to identify the active site residues of the S. scabrisporus ASNase (WP_0789718.1), three approaches were used: genomic comparison, blind molecular coupling simulation, and search for high-affinity binding pockets with AutoLigand (active site). To our knowledge, there is no information regarding the active site of the family PF06089.11 ASNases, so genomic comparison was not possible. Using AutoLigand, two possible high affinity binding sites for L-asparagine were identified (Fig. 5A). The first (site A) had a volume of 121 Å3 and an energy per volume equal to −0.2149 kcal/mol Å3; the second (site B) had a volume of 101 Å3 and an energy per volume equal to −0. 2136 kcal/mol Å3. Site A is located between an alpha-helix in the amino terminal containing the 57PRSx(2)KPxQ65 motif, and a loop in the central region of the enzyme, containing the 141NCSGKHxAML150 motif (Table 3). Site B is located in a pocket formed by a set of alpha-helices in the amino terminal of the protein, marked by the presence of the 87SHTGQxHFV95 motif. On the other hand, by performing AutoDock 4.2 whole-protein molecular coupling simulations, we found that the best ligand-enzyme interaction (L-asparagine-ASNase), with a binding free energy of −4.17 kcal/mol, targeted residues corresponding to the 141NCSGKHxAML150 motif, which correspond to the site A (Fig. 5B).

Figure 5 SsAII-2 putative binding sites.

(A) Site A (orange) contains the NCSGKHxAML sequence and site B (blue) contains the SHTGQxHFV motif. (B) Residues involved with asparagine through a direct interaction, obtained by blind molecular docking.

Additionally, in order to validate AutoLigand analysis searching active sites in the S. scabrisporus ASNases, a search for binding sites in EcAII was performed. To do this, the monomeric, dimeric, and tetrameric forms of the enzyme (the latter is the catalytically active form) were analyzed using the same conditions used for SsAII-2. It was found that AutoLigand successfully identified the binding site of L-Asn, consisting of Thr 12, Tyr 25, Ser 58, Gln 59, Thr 89, Asp 90, and Lys 162 and also Asn 248 and Glu 283 (Fig. 6), the latter two only for dimeric and tetrameric forms. The sites found (purple squares curves) had energies by volume equal to −0.2119, −0.2242, and −0.2366 kcal/mol Å3 and a volume of 136, 122, and 102 Å3 for the monomer, dimer, and tetramer, respectively (Fig. 7). It is relevant that for both the dimeric and the tetramer forms, AutoLigand successfully identified L-Asn binding pockets in EcAII: the pocket formed between the amino-terminal end of subunit A and the carboxy terminal of the subunit C, as well as equivalent pockets for dimer BD. In addition, several other solutions found by AutoLigand (curve with blue or green squares), using up to 90 filling points, converge in the different joint pockets formed by dimers.

Figure 6 EcAII dimer AutoLigand analysis.

EcAII subunit A is shown in cyan and subunit C in magenta. The red mesh represents the highest-affinity pocket found by AutoLigand (putative active site). The site represented in the scheme corresponds to the residues located at a maximum distance of 5 Å using 20 points: Thr 12, Tyr 25, Ser 58, Gln 59, Thr 89, and Asp 90 from subunit C and Asn 248 and Glu 283 from subunit A.

Figure 7 AutoLigand results for EcAII.

The minima observed in the total energy graphs per unit volume represent putative binding sites in the structures analyzed, for (A) the monomer, (B) dimer, and (C) tetramer conformation. As more filling points are used, the binding sites, cavities, or grooves are filled and the affinity decreases. The best sites are the ones with the lowest energy and the lowest volume.

Molecular docking

Molecular docking simulations were performed at the putative sites found (Table 4). For EcAII, as the reference ASNase, Thr 12, Tyr 25, Ser 58, Gln 59, Thr 89, Asp 90, Asn 248, and Glu 283 were established as flexible residues; meanwhile, molecular docking for S. scabrisporus ASNase were performed using only the rigid structure of the protein, without defining flexible side chains for L-asparagine binding.

Table 4 Molecular docking energies of ASNases.

ASNase	Free energy binding (kcal/mol)	Inter-molecular energy (kcal/mol)	Van der Waals—hydrogen bonds (kcal/mol)	Electrostatic
energy (kcal/mol)	Hydrogen bonds	
E. coli EcAII; tetramer	−9.81	−11.30	−5.88	−3.61	9	
E. coli EcAII; monomer	−8.46	−9.95	−7.02	−2.35	6	
S. scabrisporus WP_078980718.1 – site A	−6.67	−8.17	−5.25	−2.91	6	
S. scabrisporus WP_078980718.1 – site B	−4.62	−6.11	−4.39	−1.72	2	

Our results showed a higher affinity for L-asparagine of the S. scabrisporus ASNase site A than site B; however, the affinity was lower than that for EcAII. For S. scabrisporus ASNase site A, the L-asparagine interacts with residues Ser 59, Lys 62, Asn 141, Ser 143, Lys 145, His 146, Gly 237, Lys 255, and Gly 256 (Fig. 8A); for site B, the residues that interact with L-asparagine are Ala 84, Gly 78, Ser 87, Tyr 163, Leu 164, and Asp165 (Fig. 8B). Interestingly, from site A, Lys 62, Asn 141, Ser 143, Lys 145, and His 146 are highly conserved in ASNases of the PF06089.11 family.

Figure 8 Interaction maps for sites (A) and (B) from S. scabrisporus ASNase.

The black spheres represent carbon atoms, the blue nitrogen and the red oxygen. Hydrogen bonds are represented by green dotted lines and hydrophobic interactions are shown as red half-moons.

Discussion

In this work, a set of bioinformatics tools were used to identify, select, and characterize ASNases from the Streptomyces genus. ASNase identification was carried out by searching sequences homologous to EcAII and ScAII. EcAII is the best-characterized and most widely used ASNase for ALL treatment, while ScAII is a homologous ASNase related to ReAII, an atypical ASNase with no glutaminase activity and with a different immunogenic profile than EcAII (Huerta-Saquero et al., 2013). The search for homologous sequences resulted in two sets of sequences with a high probability of being ASNases (E value < 1e−06). These sequence sets, in turn, were classified into two different protein families based on their homology, using HMMer: PF00710.11 and PF06089.11, according to the classification of the PFAM database. So far, most of the reported ASNases belong to the PF00710.11 family and have been extensively studied. EcAII and the E. chrysanthemi ASNase belong to this family. On the other hand, the PF06089.11 family represents a group of atypical ASNases that remain poorly characterized. Some representative reports about these ASNases include the R. etli ASNase (Ortuño-Olea & Durán-Vargas, 2000; Moreno-Enriquez et al., 2012; Huerta-Saquero et al., 2013).

Interestingly, the BLAST results showed a greater abundance of PF06089.11 family sequences compared to the PF00710.11 family in Streptomyces. In addition, we found that about 20% of species have ASNase isoforms. In that sense, many Gram-negative bacteria have at least two isozymes of the family PF00710.11 (Fernández & Zúñiga, 2006) and, in E. coli, the existence of a third isoenzyme has been recently reported (Da Silva et al., 2018). Historically, the genus Streptomyces has been attractive due to the wide repertoire of bioactive molecules produced. However, searching for ASNases of pharmacological interest has been done only rarely.

After the identification of two sets of ASNases, we chose T-cell ED as the immunogenicity indicator, according to Cantor et al. (2011), Fernandez et al. (2014), and Galindo-Rodríguez et al. (2017), who proposed that HLA class II molecules play a critical role in the development of specific anti-ASNase antibodies and in hypersensitivity to the enzyme (Cantor et al., 2011; Fernandez et al., 2014; Galindo-Rodríguez et al., 2017). Additionally, it has been shown that proteins that are highly immunogenic generally contain a greater amount of T-cell epitopes, or clusters thereof (Singh et al., 2012). In addition, the measurement and prediction of ED have generated interest as useful tools for comparisons between therapeutic proteins, allowing selection of the best candidate in terms of probable immunogenicity (De Groot & Martin, 2009). In this sense, our results showed that ASNases of the PF06089.11 family contain lower EDs than enzymes of the PF00710.11 family, as well as fewer epitope clusters throughout the sequence. In addition, the allele coverage, which is related to the percentage of the population that develops a significant immune response, showed Streptomyces ASNases to be potential pharmacological options. In other words, due to their low content of T-cell epitopes, low antigenicity profile, and low allele coverage, Streptomyces ASNases represent, in terms of immunogenicity, a pharmacological alternative for ALL treatment. In this sense, the Streptomyces brollosae NEAE-115 ASNase has better cytotoxicity and immunogenicity profiles for use in ALL treatment, based on evaluation in a murine model, compared with EcAII (El-naggar et al., 2018). Previously, anticancer activity of the Streptomyces fradiae NEAE-82 ASNase in colon cancer cell cultures was reported (El-Naggar et al., 2016).

For the PF06089.11 family of ASNases, the lack of information of the active site precludes direct comparison, as was used in the approach for the ASNase WP_078979039.1. However, the use of computational tools based on structure inspection and on the evaluation of affinity maps has proven highly effective in identifying probable binding sites in uncharacterized proteins (Harris, Olson & Goodsell, 2008). Based on the use of computational tools, it was possible to identify two putative binding sites in SsAII-2 (WP_078980718.1). Interestingly, in both sites the motifs NCSGKHxAM, PRSx(2)KPxQ, and SHTGQx(2)H were identified, and these motifs are highly conserved in the PF06089.11 family (Moreno-Enriquez et al., 2012). Of these, Borek & Jaskólski (2001) proposed that some of the residues of the NCSGKHxAM motif could be involved in the hydrolytic deamidation of L-asparagine.

On the other hand, the residues we found conserved in this family of asparaginases resemble those of the active site of the Ntn amidotransferases, in which, among the important residues for glutamine deamidation are found Cys, Asn, and Gly, all of them present in NCSGKHxAM motif, and the deamidation mechanism proceeds with an oxyanion formation with the substrate. Although this mechanism is described for glutamine amidohydrolases, it may be a mechanism similar to that of this family of asparaginases, whose active site is different from those of the PF00710.11 family (Isupov et al., 1996).

Although site A showed higher affinity for L-asparagine binding, additional studies are needed to confirm the best site for ligand binding. Additionally, molecular dynamics simulations can provide more evidence of the characteristics of the binding site and, together with in vitro studies, will be useful for understanding the mechanism of enzymatic reaction (Karplus & Kuriyan, 2005). Although our results predicted that SsAII-2 has a lower affinity than EcAII, its different folding and immunogenic characteristics place it as a good candidate. Identifying catalytic site residues will allow us to perform site-directed modifications to increase its affinity.

The strategy developed here can be applied to the search for asparaginases in other clades of microorganisms, and even in eukaryotes, specifically mammalian asparaginases, whose evolutionary proximity to humans predicts less immunogenicity.

Conclusions

In summary, the search for ASNases in phylogenetically distant microorganisms and the application of bioinformatic tools to assess their toxicity and affinity for L-asparagine are viable approaches to obtain new ASNases with therapeutic potential. Based on its low immunogenicity and excellent enzymatic activity predicted, we have identified the S. scabrisporus ASNase as a potential alternative for the treatment of ALL. The subsequent enzymatic and immunogenic characterization of the S. scabrisporus ASNase is necessary for the validation of this bioinformatic approach.

We acknowledge Katrin Quester and Itandehui Betanzo for technical assistance.

Additional Information and Declarations

Competing Interests

Author Contributions

Data Availability

The authors declare that they have no competing interests.

Iván González-Torres conceived and designed the experiments, performed the experiments, analyzed the data, prepared figures and/or tables, authored or reviewed drafts of the paper, and approved the final draft.

Ernesto Perez-Rueda conceived and designed the experiments, performed the experiments, analyzed the data, prepared figures and/or tables, authored or reviewed drafts of the paper, and approved the final draft.

Zahaed Evangelista-Martínez performed the experiments, authored or reviewed drafts of the paper, and approved the final draft.

Andrés Zárate-Romero performed the experiments, authored or reviewed drafts of the paper, and approved the final draft.

Angélica Moreno-Enríquez analyzed the data, authored or reviewed drafts of the paper, and approved the final draft.

Alejandro Huerta-Saquero conceived and designed the experiments, analyzed the data, authored or reviewed drafts of the paper, and approved the final draft.

The following information was supplied regarding data availability:

The data is available in Tables 1–4.

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
