# Peer review of "Identification of L-asparaginases from Streptomyces strains with competitive activity and immunogenic profiles: a bioinformatic approach"

_PeerJ, doi:10.7717/peerj.10276_

## Round 0.1 · original submission · Major Revisions

Dear Dr. González-Torres and colleagues:

Thanks for submitting your manuscript to PeerJ. I have now received three independent reviews of your work, and as you will see, two of the reviewers raised some concerns about the research (and manuscript). Despite this, these reviewers are optimistic about your work and the potential impact it will have on research studying applied clinical aspects of L-asparaginases. Thus, I encourage you to revise your manuscript, accordingly, taking into account all of the concerns raised by all three reviewers.

Please address the robustness issues raised by the reviewers; your literature searches seemed limited and you stand to do well from considering a wider range of organisms. I am less concerned about conducting laboratory work, but a detailed plan about how such work will (or should by others) be carried out.

As pointed out by Reviewer 2, your bioinformatic approach would benefit from being able to distinguish between a low Km (clinically relevant) and a high Km (clinically irrelevant) ASNase. Please focus on this in your revision.

Therefore, I am recommending that you revise your manuscript, accordingly, taking into account all of the issues raised by the reviewers.

Good luck with your revision,

-joe

Reviewer 1 ·

Basic reporting

Good

Experimental design

Good

Validity of the findings

Good

Additional comments

Further genetic engineering studies on L-aspa is highly appreciable!

·

Basic reporting

The manuscript by Gonzales-Torres et al addresses an important need, namely, that of identifying a new asparaginase (ASNase) that would have clinical advantages over the currently approved ASNases (which are anti-leukemic biologic drugs). However, the work has major deficiencies that detract from its impact.

This same group proposed in 2012 that an ASNase from Rhizobium etli (ReAII) has ‘therapeutic potential’ (Lines 84-88). This assertion is perplexing since according to the 2012 paper, ReAII has a Km value 600-fold higher than that of EcAII (Table 2 of the 2012 paper) – see point #4 below of ‘Major Issues’ section. In line 214 the authors write that ‘ReAII hydrolyzes L-asparagine as similar levels to EcAII, but with lower activity’ (and site the 2012 paper). This statement gives this reviewer pause, since it demonstrates a total misunderstanding of what type of activity is required for clinical efficacy. We need to care less about the rate of Asn hydrolysis when [Asn] is in the millimolar range. The only relevant number is the rate of Asn hydrolysis at physiological [Asn], which is 50 micromolar.

Major issues
1. The authors fail to be explicit about what properties would make an asparaginase a suitable anti-cancer drug. They note, correctly, the importance of identifying new ASNases that are ‘less toxic and less immunogenic’ (line 90). However, the authors throughout the manuscript fail the mention the concept of ‘Km value’ and that it is imperative for the ASNase to have a Km value in the low micromolar range.
2. When searching for less immunogenic ASNases, it is puzzling that the authors do not discuss searching mammalian genomes. What is the rationale for focusing on yet another bacterium, when the current ASNases are bacterial and are highly immunogenic? Why not include mammalian ASNases in the search, since, one can make the assumption, that a more human-like ASNase would likely be less immunogenic than a more distant ASNase.
3. E. coli has several ASNases. The clinically used one, EcAII is structurally very similar to EcAI. However, whereas EcAII has the required low Asn Km value, EcAI does not. Therefore, any bioinformatic search for a new ASNase must be able to distinguish between a low Km (clinically relevant) and a high Km (clinically irrelevant) ASNase. The authors need to include a section on how the bioinformatic tools that they use make this differentiation (if they do). If the tools do NOT make this differentiation, that this work is largely not helpful since it misses the top criterium for an efficient ASNase. If the authors believe that by adding EcAI to the phylogenetic analysis this was actually done, they are asked to be more explicit about it. Also, please discuss the presence of a periplasmic secretion sequence in EcAII but not in EcAI, and whether this can act as a differentiator between Type I and Type II ASNases (and as such, predict ASNases with low Km value). The authors hint at this (Line 198), but if so, it needs more explicit discussion.
4. As discussed above, Km is the most important parameter for clinical relevance. It is unclear why the authors include ReAII in their analysis since it has a Km that is way too high.
5. The authors write that ASNases with sequences homologous to that of ScAII (S. coelicolor type II) are in the same family as that of ReAII. In other words, those ASNases in family PF06089.11 are, to my understanding, not suitable for clinical use based on a predicted high Asn Km value. So why are these even considered in this work?
6. The authors find that the ReAII family (PF06089.11) is predicted to be less immunogenic than the EcAII family (PF00710.11). Assuming this is correct, it does not suggest that PF06089.11 family members be tested since they would be predicted to have a high Km, and hence, clinically, irrelevant.
7. This reviewer sees no value in the protein structure predictions. Can these predictions say something about the Km value? I think not. If the authors disagree, please explain how these data would inform the selection of an ASNase to test.
8. At the end of the day, the proof of the pudding is in the eating. In other words, the authors need to pick the ‘best’ ASNase based on the predictions of all of the programs that they ran, express it, and confirm that it has the required low Km value. Without such confirmatory work, the value of this report is deemed low.

Minor issues
1. Line 51: there are actually 4, not 3 ASNase formulations currently available. A new PEGylated EcA formulation, which is different to Oncaspar by only having a different linker between enzyme and PEG, has been recently approved.
2. Line 53: In the US, Elspar (EcAII) is not used anymore, and Oncaspar (EcAII-PEG) has become the first line treatment.
3. Line 61: please look into liver toxicity as well.
4. Line 71: please explain why you connect the glutaminase co-activity with allergic reactions. I can understand why the bacterial origin would cause an allergic reaction, but why do you include the glutaminase activity. Of course, the glutaminase activity is likely to be behind many of the other toxicities seen with asparaginases (but not immunogenicity).

Grammar/English suggestions
1. The word ‘Selection’ in the title is misleading since it implies some sort of an experimental selection assay. A more appropriate word would be ‘Identification’.
2. Line 27; move ‘predicted’ to the front; i.e. ‘Based on it predicted low immunogenicity…..’
3. Line 69; these ‘enzymes’ should be these ‘antibodies’
4. Line 84; typo; ASNasa  ASNase
5. Line 138; DRB1*11:01 is listed twice.

Experimental design

Included above

Validity of the findings

Included above

Reviewer 3 ·

Basic reporting

The authors use appropriate concise language and provide a relevant introduction discussion that describes the current state of the field and limitations to current asparaginase therapy during the treatment of leukemias. The figure legends and labels are appropriate and clear.

Experimental design

The authors use sophisticated methods for their research objective aiming to identify a new asparaginase source that can be less immunogenic relative to currently used preparations. However, there are a few areas of opportunity to clarify their approach using the Immune Epitope Database (IEDB). In particular, with regards to the following:

1. For epitope density (ED) estimation, the equation used includes a parameter “Predicted epitope” which is not clear. Is this the CPR of the predicted epitope? If so, it should be clarified.

2. For Table 2:
a. The authors should clarify whether “epitope number” refers to the number of epitopes with CPR < 1.
b. For HLA “Allele number” on table 2, the authors should clarify whether this refers to the # of HLA alleles (out of 8) where that particular epitope has a CPR < 1.

Generally, the antigenicity/HLA binding data is used clearly to compare to E. coli asparaginase (EcAII). The manuscript would benefit by including the same analysis for Erwnia asparaginase (ErAII), as both preparations are used clinically. Furthermore, demonstrating that epitopes of EcAII and ErAII don’t cross-react with asparaginase from Streptomyces scabrisporus is key for recommending it as a therapeutic option after immune responses to both agents (not clear from Fig. 3).

Validity of the findings

The authors’ conclusions are supported by their results. However, an important limitation of the study is that the stability of asparaginase from Streptomyces scabrisporus in plasma or serum was not assessed or estimated. The authors should acknowledge importance of PK properties/clearance for identifying a suitable alternative asparaginase preparation that can be used clinically. Note that an asparaginase with high clearance will have low immunogenicity due to decreased immune cell presentation.

Additional comments

The manuscript by Gonzales -Torres et al. presents a computational approach for identifying less immunogenic forms of the chemotherapeutic agent asparaginase. The authors use a novel approach that screens candidates for antigenicity and binding to MHC class II alleles, which is an important consideration for therapeutic asparaginases. Based on their search and approach, they identify a previously uncharacterized L-asparaginase from Streptomyces scabrisporus as a promising therapeutic candidate that can be used after human immune responses to PEGylated E. coli asparaginase and/or Erwinia asparaginase.

The authors focus on an important research question related to asparaginase therapy, which is an essential component of pediatric leukemia treatment protocols worldwide. Their approach using bioinformatic tools to assess whether a particular asparaginase would have a high probability of a human immune response is novel and provides a new tool or strategy for screening potential therapeutic enzymes. The strength of the manuscript is the approach used for identifying their proposed asparaginase. The most important issue is providing a more detailed discussion and conclusion on whether Streptomyces scabrisporus asparaginase would cross-react after E. coli or Erwinia asparaginase.

---

## Round 0.2 · Minor Revisions

Dear Dr. González-Torres and colleagues:

Thanks for revising your manuscript. The reviewers are very satisfied with your revision (as am I). Great! However, there are a few minor edits to make. Please address these ASAP so we may move towards acceptance of your work.

Best,

-joe

Reviewer 1 ·

Basic reporting

Good

Experimental design

Good

Validity of the findings

Good

Additional comments

Further molecular characterization of L-asparaginase is highly appreciable

·

Basic reporting

The author’s revised manuscript is definitely improved over the original submission. Yet, some issues that need attention remain.

Major required change:
Section 2.7. Here the authors discuss the molecular docking results for S. scabrisporus ASNase (SsAII), with the goal of identifying the active site. The result is shown in Figure 8. From this figure, it is not clear how this enzyme would catalyze the hydrolysis of asparagine. Specifically, for the predicted Site A, the amide group of the asparagine side chain is predicted to be close to Gly237; the carbonyl to Asn141; neither of these amino acids are likely to participate in catalysis. The predicted binding to Site B also fails to show the type of amino acids that could catalyze the hydrolysis reaction.

For example, for the EcA ASNase, conserved threonine residues, a tyrosine, and a lysine participate in catalysis. The authors need to re-examine the predicted binding of asparagine to SsAII to see if it is consistent with being near to residues that can catalyze the reaction. And are the identified putative catalytic residues conserved in this family of ASNases? If not, then this prediction is likely to be false.

Minor required changes:
Title: Identification of L-asparaginases from Streptomyces strains with competitive activity and immunogenic profiles: a bioinformatic approach  Identification of L-asparaginases from Streptomyces strains with a predicted competitive activity and immunogenic profiles: a bioinformatic approach

Abstract: In this work, a bioinformatic approach was used to identify, select and characterize L-asparaginases from Streptomyces through sequence…. In this work, a bioinformatic approach was used to identify, select and computationally characterize L-asparaginases from Streptomyces through sequence….

Page 3:…as well as pegylated E. chrysanthemi ASNase (EcAI-PEG).  as well as pegylated E. chrysanthemi ASNase (ErAI-PEG).

Experimental design

see above

Validity of the findings

see above

Reviewer 3 ·

Basic reporting

See previous comments

Experimental design

See previous comments

Validity of the findings

See previous comments

Additional comments

Thanks for adding clarifications.

---

## Round 0.3 · Minor Revisions

Dear Drs. Gutiérrez-Fonseca and Ramírez:

Thanks for revising your manuscript. It mostly looks great except for one concern. In looking over your rebuttal, it is not clear to me that you addressed a specific issue raised by Reviewer 2. Ideally, your claims regarding similarity between their active sites and those of Ntn amiditransferases should be supported by comparison with the 1XFG crystal structure. Please address this so we can move forward with accepting your work for publication.

Best,

-joe

---

## Round 0.4 · Minor Revisions

Dear Dr. González-Torres and colleagues:

Thanks for revising your manuscript. However, your structural analysis is not convincing at all. Please (at least) provide the coordinates of your model and a figure comparing your proposed active site with that of the 1XFG crystal structure. We have also noticed that section 2.5 lacks any data regarding the actual results returned by RAMPAGE and verify3D, the C-scores of their models (and how that scores compares with the typical values for good models) etc. You cannot simultaneously claim "resemble those of the active site of the Ntn amidotransferases, in which, among the important residues for glutamine deamidation are found Cys, Asn, and Gly, and the deamidation mechanism proceeds with an oxyanion formation with the substrate. " and provide such a paucity of confirming data.

Please address this as described above.

Best,

-joe

---

## Round 0.5 · Minor Revisions

Dear Dr. González-Torres and colleagues:

Thanks for revising your manuscript. However, your structural analysis is still not convincing, and it raises some doubt with your interpretations. Your "structural alignment" in the Supporting information shows that the fold of your protein does not resemble that of the glutaminase at all. This might not be too much of a problem if the putative active site cavity were similar, the region of the putative active site in Fig, 9, (which you decided to show as ribbons instead of actual atoms) is also extremely different, but that doesn't prevent you from claiming similarity.

At this time, I do not feel comfortable with accepting your work in its current form. I believe the modeling does not warrant you interpretations at all. Please considering revising your work and aborting framing your conclusions based on this comparative analysis.

Best,

-joe

---

## Round 0.6 · accepted · Accept

Dear Dr. González-Torres and colleagues:

Thanks for revising your manuscript based on the concerns that were raised. I now believe that your manuscript is suitable for publication. Congratulations! I look forward to seeing this work in print, and I anticipate it being an important resource for groups studying applied clinical aspects of L-asparaginases. Thanks again for choosing PeerJ to publish such important work.

Best,

-joe